# Correlation of Immune-Inflammatory Markers with Clinical Features and Novel Location-Specific Nomograms for Short-Term Outcomes in Patients with Intracerebral Hemorrhage

**DOI:** 10.3390/diagnostics12030622

**Published:** 2022-03-02

**Authors:** Hsien-Ta Hsu, Pei-Ya Chen, I-Shiang Tzeng, Po-Jen Hsu, Shinn-Kuang Lin

**Affiliations:** 1Division of Neurosurgery, Taipei Tzu Chi Hospital, Buddhist Tzu Chi Medical Foundation, New Taipei City 23142, Taiwan; j1208192@ms45.hinet.net; 2School of Medicine, Tzu Chi University, Hualien 97004, Taiwan; ruentw@gmail.com; 3Stroke Center and Department of Neurology, Taipei Tzu Chi Hospital, Buddhist Tzu Chi Medical Foundation, New Taipei City 23142, Taiwan; b101091026@tmu.edu.tw; 4Department of Research, Taipei Tzu Chi Hospital, Buddhist Tzu Chi Medical Foundation, New Taipei City 23142, Taiwan; istzeng@gmail.com

**Keywords:** Glasgow Coma Scale, immune-inflammatory marker, intracerebral hemorrhage, intracerebral hemorrhage score, nomogram, unfavorable outcome, mortality

## Abstract

(1) Background: We investigated the association of four immune-inflammatory markers with clinical features and established location-specific nomograms to predict mortality risk in patients with intracerebral hemorrhage (ICH). (2) Methods: We retrospectively enrolled 613 inpatients with acute ICH. (3) Results: Overall mortality was 22%, which was highest in pontine hemorrhage and lowest in thalamic hemorrhage. All four immune-inflammatory markers exhibited a positive linear correlation with glucose, ICH volume, ICH score, and discharge Modified Rankin Scale (mRS) score. Significant predictors of death due to lobar/putaminal hemorrhage were age, glucose and creatinine levels, initial Glasgow Coma Scale (GCS) score, ICH volume, and presence of intraventricular hemorrhage. None of the immune-inflammatory markers were significant predictors of unfavorable outcome or death. We selected significant factors to establish nomograms for predicting death due to lobar/putaminal, thalamic, pontine, and cerebellar hemorrhages. The C-statistic for predicting death in model I (comprising factors in the establishment of the nomogram) in each type of ICH was higher than that in model II (comprising ICH score alone), except for cerebellar hemorrhage. These nomograms for predicting death had good discrimination (C-index: 0.889 to 0.975) and prediction probabilities (C-index: 0.890 to 0.965). (4) Conclusions: Higher immune-inflammatory markers were associated with larger ICH volume, worse initial GCS, and unfavorable outcomes, but were not independent prognostic predictors. The location-specific nomograms provided novel and accurate models for predicting mortality risk.

## 1. Introduction

Hemorrhagic stroke accounts for 8–15% of strokes in Western countries and 18–24% in Asian countries and carries a much higher risk of mortality than ischemic stroke does [1,2,3,4]. Immune-inflammatory processes are involved in all stages of acute stroke, including ischemic and hemorrhagic strokes [5,6,7,8]. Innate immunity—the immune response present at birth—is mainly provided by neutrophils, monocytes, macrophages, natural killer cells, and complement systems. Adaptive (or acquired) immunity is provided by lymphocytes, which deliver antigen-dependent and antigen-specific responses to invasion [9]. The white blood cell (WBC) count, neutrophil count (NC), neutrophil-to-lymphocyte ratio (NLR), and systemic immune-inflammation index (SII) represent the status of an inflammatory response and are regarded as immune-inflammatory markers; their high levels are independent risk factors for poor outcomes in patients with acute ischemic stroke [10,11,12]. However, the association of immune-inflammatory marker levels with the outcomes of patients with acute hemorrhagic stroke remains controversial [7,13,14,15].

Various scores have been established for quickly assessing the outcomes of patients with intracerebral hemorrhage (ICH), such as the ICH score, ICH grading scale, modified ICH score, new modified ICH score, simplified ICH score, ICH index, and modified emergency department ICH grading scale [16,17,18,19,20,21,22]. Most of the score scales for ICH assessment are based on age, level of consciousness, and the location and size of the hemorrhage. ICH volume plays a major role in the scoring system. However, the ICH location also carries a critical risk of an unfavorable outcome. Dividing the ICH location into merely supratentorial and infratentorial is inadequate for weighing the influence of outcomes. In 2016, the Taiwan Joint Commission on Hospital Accreditation validated the ICH score for the assessment of initial severity of patients with ICH as early as possible after admission to the hospital to ensure the quality of care. The ICH score, established by Hemphill et al. in 2001, has been widely used to assess stroke severity and outcomes in patients with ICH [16]. Hemphill et al. suggested that the ICH score can help with risk stratification for ICH treatment studies but not as a precise predictor of outcomes.

Here, we investigated the association of the four immune-inflammatory markers—WBC count, NC, NLR, and SII—with clinical features and the outcomes of patients with ICH. We also sought to determine the appropriate criteria for different hematoma locations compared with the ICH score in predicting the outcomes of patients with ICH and to establish nomograms to predict the probabilities of mortality at different ICH locations.

## 2. Materials and Methods

### 2.1. Stroke Population and Data Collection

The stroke registry database was retrospectively reviewed to identify patients who received stroke treatment for spontaneous ICH in a neurosurgical ward from January 2016 to October 2021. The inclusion criteria were (1) a diagnosis of acute ICH confirmed with clinical presentations and (2) evidence of hemorrhagic lesions according to brain computed tomography (CT) or magnetic resonance imaging. Patients with traumatic ICH, tumor bleeding, or ICH due to arteriovenous malformation were excluded. Information on sex; age; risk factors, namely, history of hypertension, diabetes mellitus, heart disease, and prior stroke; and length of hospital stay were recorded for analysis. Laboratory data obtained on arrival at the emergency department included complete blood count with WBC differentials, platelet count, C-reactive protein (CRP), and glucose and creatinine levels, which were measured in the hospital central laboratory using an XN-9000 Compact Integration (Sysmex Corporation, Kobe, Japan) on receipt of the sample. The immune-inflammatory markers were obtained from retrospective calculation: NLR was calculated as the ratio of NC to lymphocyte count. SII was calculated as platelet count multiplied by NLR.

### 2.2. Ethics Statement 

The study was conducted in accordance with the Declaration of Helsinki. Ethical approval for this study was provided by the Institutional Review Board of Taipei Tzu Chi Hospital, New Taipei City (approval no. 10-XD-156). The requirement for informed written consent was waived because the study was a retrospective data analysis. All data collected and analyzed in this retrospective study were derived from clinical records without any intervention or influence on clinical treatment.

### 2.3. Stroke Severity and Clinical Features

The ICH score was used by neurosurgeons to assess the stroke severity within 6–12 h of patient arrival at the emergency department [16]. The ICH score comprises five items: Galsgow Coma Scale (GCS) score (13–15 = 0, 5–12 = 1, 3–4 = 2), ICH volume (≥30 cm^3^ = 1, <30 cm^3^ = 0), intraventricular hemorrhage (IVH) (presence = 1, absence = 0), infratentorial origin of ICH (presence = 1, absence = 0), and age (≥80 years = 1, <80 years = 0). The total ICH score ranges from 0 to 6, with higher scores indicating greater stroke severity. Brain CT was performed with 4-mm-thick slices. During the retrospective review, the ICH volume was measured again on the initial brain CT scans by a stroke neurologist using the *ABC*/2 method, where *A* is the greatest diameter on the largest hemorrhage slice, *B* is the diameter perpendicular to *A*, and *C* is the greatest diameter on the largest hemorrhage slice in sagittal view [23]. Functional outcomes were evaluated at discharge by neurosurgeons using the Modified Rankin Scale (mRS; a grading scale scoring 0 to 6 with 1-point increments; the higher the score, the more severe the disability; a grade 0 indicates no symptoms and a grade 6 indicates death). Given that patients with hemorrhagic stroke might have worse short-term outcomes than those with ischemic stroke, an mRS score > 3 was considered to indicate an unfavorable outcome. Predictors of unfavorable outcome (mRS score > 3) and death (mRS score = 6) were analyzed according to different hemorrhage locations.

### 2.4. Identification of Hemorrhage Location

The locations of hemorrhage were categorized into supratentorial, infratentorial, and intraventricular ICH. Supratentorial ICH was further stratified into lobar, putaminal, thalamic, and caudate ICH, whereas the infratentorial ICH was stratified into pontine and cerebellar ICH. An overlapped hematoma was categorized as lobar, putaminal, or thalamic hemorrhage according to the location of the main part of the hematoma. Most brainstem hemorrhages were located at the pons and simply categorized as pontine hemorrhage.

### 2.5. Statistical Analysis

Continuous variables are presented as median (1st–3rd quartile). The chi-square test or Fisher’s exact test were used to compare categorical variables. Intergroup comparisons of continuous variables were performed using the Mann–Whitney *U* test or Kruskal–Wallis test as appropriate. Comparisons of the significance between the immune-inflammatory markers were done using the area under the curve (AUC) through receiver operating characteristic (ROC) curve analyses for death. A multiple or stepwise logistic regression model was used to identify significant factors associated with unfavorable outcomes and death. For better identification of location-specific characteristics of ICH, we stratified ICH into four subgroups: lobar/putaminal, thalamic, pontine, and cerebellar hemorrhage. Location-specific nomograms for predicting death in patients with one of the four main ICH types were established from logistic regression analyses of continuous variables. In addition, we compared the predictive performance of the variables by using the C-statistic for unfavorable outcomes and death between the two models. Model I comprised factors selected in the establishment of nomograms for each ICH location. Model II comprised the ICH score alone. A *p*-value < 0.05 was considered to indicate a significant result. All statistical analyses were performed using SPSS (version 24; SPSS, Chicago, IL, USA). The ROC curves were compared using MedCalc version 18 (MedCalc Software, Mariakerke, Belgium). Nomograms were developed using STATA version 17 (StataCorp, College Station, TX, USA). Validation and calibration of nomograms were performed using Orange version 3.28 [24].

## 3. Results

### 3.1. Clinical Features of Patients with Intracerebral Hemorrhage

During the study period, 613 patients with acute ICH were enrolled. Table 1 summarizes the clinical features of 613 patients with ICH stratified by gender. There were 395 men and 218 women with a median age of 64 years. The median WBC count, NC, NLR, and SII were 9.0 × 10^3^/mL, 6.1 × 10^3^/mL, 3.5, and 746, respectively. The CRP value was available only in 57% (351/613) of patients and was not included in further analysis to avoid bias. The median initial GCS score and discharge mRS score were 13 and 4, respectively. Unfavorable outcomes (mRS > 3) were observed in 69% of patients and death occurred in 22% of patients. Women were older and had a higher level of platelet counts and glucose, while men had higher levels of diastolic blood pressure, hemoglobin, and creatinine.

### 3.2. Distribution of Intracerebral Hemorrhage

There were 495 patients that had supratentorial ICH, 106 patients had infratentorial ICH, and 12 patients had IVH. Among the patients with supratentorial ICH, 238 had putaminal, 136 had lobar, 113 had thalamic, and 8 had caudate hemorrhages. Among the patients with infratentorial ICH, 59 had pontine and 47 had cerebellar hemorrhages. Table 2 summarizes the clinical features of all patients stratified by ICH location. Patients with caudate hemorrhage and IVH were not included in the subsequent comparisons owing to their small sample size and incalculable ICH volume. Among the remaining patients, those with lobar hemorrhage were the oldest and had the lowest hemoglobin levels. Patients with cerebellar ICH had the highest WBC, NLR, and glucose levels. The median ICH volumes were 23 cm^3^, 14 cm^3^, 4 cm^3^, 2 cm^3^, and 9 cm^3^ in lobar, putaminal, thalamic, pontine, and cerebellar hemorrhages, respectively. Except for patients with only IVH, IVH was also observed in all patients with caudate ICH. The mortality rate in all 613 patients with ICH was 22%. Patients with pontine hemorrhage had the lowest initial GCS score (11) and the highest mortality rate (36%), whereas patients with pontine and cerebellar hemorrhage had the highest ICH score (2). Patients with thalamic hemorrhage had the lowest mortality rate (11%). No differences in discharge mRS score and unfavorable outcomes were observed.

### 3.3. Correlation of Immune-Inflammatory Markers with Clinical Features

Table 3 presents the correlation of the four immune-inflammatory markers with age, variables, and clinical features in all patients. Age had a negative linear correlation with WBC count, NC, hemoglobin, platelet count, and initial GCS score and a positive linear correlation with ICH volume, ICH score, and discharge mRS score. All four immune-inflammatory markers exhibited a positive linear correlation with glucose level, ICH volume, ICH score, and discharge mRS score and a negative linear correlation with initial GCS score. Furthermore, the ICH score also exhibited a positive linear correlation with glucose level (*r*^2^ = 0.055, *p* < 0.001) and discharge mRS score (*r*^2^ = 0.417, *p* < 0.001).

After 20 patients with caudate hemorrhage or IVH were excluded, 593 patients were stratified into four groups: lobar/putaminal, thalamic, pontine, and cerebellar hemorrhage. Table 4 presents the univariate and multivariable analyses of clinical features and outcomes in 374 patients with lobar/putaminal hemorrhage. Patients with unfavorable outcomes and death were older; had lower hemoglobin levels and initial GCS scores; and had higher WBC counts, NC, NLR, glucose and creatinine levels, ICH volumes, ICH scores, and presence of IVH. Patients with unfavorable outcomes had longer hospital stays and a higher rate of prior stroke, whereas patients with death had shorter hospital stays. No difference in SII was observed. Through ROC curve analysis, we identified cutoff points for the prediction of death for WBC, NC, and NLR as >9.6 × 10^3^/mL (AUC = 0.690), >8.2 × 10^3^/mL (AUC = 0.656), and >4.6 (AUC = 0.592), respectively. WBC count exhibited the most significant difference among the four immune-inflammatory markers and was selected for further multivariable analysis.

### 3.4. Factors Influencing Unfavorable Outcomes and Death and Development of Nomograms

Significant predictors through multiple logistic regression analysis were initial GCS score, ICH volume, and presence of IVH for unfavorable outcomes (not shown in Table 4); and were initial GCS score, ICH volume, presence of IVH, age, and creatinine for death (*p* < 0.05) (Table 4). On the basis of the results of multivariable analyses, we chose factors with a *p*-value of <0.1 (thus further including glucose) to establish a nomogram for predicting death in patients with lobar/putaminal hemorrhage (Figure 1A). We selected the effect of the chosen factors in the development of nomogram as model I and selected ICH score alone as model II. The predictors in models I and II for unfavorable outcome and death were further analyzed using the C-statistic to compare the predictive performance. The C-statistics in lobar/putaminal hemorrhage for predicting unfavorable outcomes and death in model I were 0.876 and 0.931, respectively, and were higher than those of model II (*p* < 0.05) (Table 5). Using the same univariate and stepwise logistic regression, we analyzed the significant predictors of unfavorable outcome and death (Appendix A) and established nomograms for predicting death due to thalamic, pontine, and cerebellar hemorrhages (Figure 1B–D). ICH volume was the only significant predictor of death due to thalamic hemorrhage, whereas initial GCS score was the only significant predictor of death due to pontine and cerebellar hemorrhage.

However, considering the possible bias due to a smaller number of patient groups, we still selected both initial GCS score and ICH volume, which were listed as dominant factors in the ICH score, as important predictors of death to establish nomograms. Age was also enrolled in the establishment of nomogram in cerebellar hemorrhage due to a *p*-value of <0.1. Table 5 presents the results of the C-statistics of models I and II for unfavorable outcomes and death for different hemorrhage locations. Model I exhibited a higher predictive performance for both unfavorable outcomes and death due to lobar/putaminal and thalamic hemorrhage, and higher predictive performance for death due to pontine hemorrhage (*p* < 0.05). No difference in predictive performance was observed between models I and II for both unfavorable outcomes and death due to cerebellar hemorrhage and for unfavorable outcomes from pontine hemorrhage. The area under the ROC curve, or C-index, in internal validation with 5-fold cross-validation of the nomograms ranged from 0.889 to 0.975 for each type of ICH (Figure 2A–D). Further calibration plots indicated that the prediction probability for each type of ICH ranged from 0.778 to 0.900, while the classification accuracy ranged from 0.890 to 0.965 (Figure 2E–H).

## 4. Discussion

In our cohort, lobar and putaminal hemorrhage comprised 61% of all intracerebral hemorrhages. Nearly 70% of all patients had unfavorable outcomes. Mortality was most likely from pontine hemorrhage (36%) and least likely from thalamic hemorrhage (11%). The median size of the hematomas was largest in lobar hemorrhages (23 cm^3^), followed by putaminal (14 cm^3^) and cerebellar (9 cm^3^) hemorrhages. Patients with pontine hemorrhages had the smallest median size of hematomas but the lowest initial GCS. The median hospital LOS was only 4 days in patients with a fatal outcome.

Immune-inflammatory markers are significant predictors of unfavorable outcomes in patients due to acute ischemic stroke [10,12,25]. The median age of onset for acute hemorrhagic stroke was 64.0 years, which was lower than that for acute ischemic stroke (72.1 years) [26]. An atherosclerotic process, characterized by chronic inflammation that causes large and medium arterial thromboses, progresses with age and is, therefore, more prominent in older adults [27]. Thus, chronic inflammation occurs more in patients with acute ischemic stroke than in patients with acute ICH before stroke onset. The inflammatory response during acute ischemic stroke activates the immune system. Innate immunity, involving mainly neutrophils, is rapidly activated, followed by activation of the adaptive immunity involving lymphocytes. During an acute hemorrhagic stroke, the inflammatory response is also immediately triggered by hematoma components [14]. Neutrophils are the first WBCs to enter the hematoma bed after an ICH onset [28]. They cause neurotoxicity by increasing capillary permeability and breaking down the blood–brain barrier, which enhances hematoma growth and cerebral edema [29]. The average WBC counts in patients with ICH in the present study was 9.8 × 10^3^/mL (median 9.0 × 10^3^/mL), which were higher than the reported value of 8.0 × 10^3^/mL (median 7.5 × 10^3^/mL) in our previous study involving patients with acute ischemic stroke [26]. The average and median NLR were also higher in patients with ICH (6.3 and 3.5, respectively) than in those with ischemic stroke (4.5 and 2.8, respectively) [26]. The inflammatory response may occur more in hemorrhage than in ischemia during acute stroke. All four immune-inflammatory markers had significant positive correlations with hematoma size and initial stroke severity, including initial GCS and ICH scores. However, none of the four immune-inflammatory markers was significantly associated with unfavorable outcomes or death in the multivariable analysis. This result is consistent with several studies, particularly when the initial GCS score and ICH volume, which were the strong predictors, were adjusted in multivariable analyses [7,13,14,15]. Hence, although the levels of immune-inflammatory markers were higher in acute hemorrhagic stroke compared with acute ischemic stroke, they were not independent risk factors for unfavorable outcomes or death. Therefore, we did not enroll immune-inflammatory markers into the establishment of nomograms for the prediction of death.

Hematoma size and ICH location critically influence stroke severity and outcomes. Indeed, hematoma ≥ 30 cm^3^ and infratentorial origin are each assigned 1 point on the ICH score. In the present study, the median ICH volume of lobar hemorrhage (23 cm^3^) was larger than that of putaminal hemorrhage (14 cm^3^). Nevertheless, mortality was much higher in patients with putaminal hemorrhage (26% vs. 20%), even though patients with lobar hemorrhage were older. Thalamic and lobar hemorrhage were categorized in the same supratentorial group. However, a hematoma of 23 cm^3^, which might not be dangerous when located at the subcortical area but can be life threatening if located at the thalamus, is assigned as having the same ICH score of 0 points. Similarly, a hematoma of 11 cm^3^, which is relatively innocuous when located at the cerebellar hemisphere but usually causes altered consciousness and quadriplegia if located at the pons, is given as the same score of 1 point for its infratentorial origin. Given the complexity of the impact of ICH volume and location on the outcome, evaluating the affecting factors separately is preferable for different ICH locations. In acute ischemic stroke, recognizing the actual location and volume of the infarct area is usually infeasible on initial brain CT and mostly relies on follow-up magnetic resonance imaging. In contrast, hematomas are clearly displayed as hyperdense lesions, allowing for easy discernment of their location and volume on an initial CT. Most hematomas can be stratified into lobar/putaminal, thalamic, pontine, or cerebellar hemorrhage according to the location of the main part of the hematoma. Together with the GCS evaluation, we were able to predict the clinical outcome more accurately with nomograms for four ICH location groups. IVH, which comprised approximately 2% of all ICH in the present study, is not included in most reported assessment systems because of incalculable ICH volumes and unclassifiable supratentorial or infratentorial location. Caudate hemorrhages, which comprised only 1% of all ICHs, usually complicated IVH with incalculable ICH volumes.

Age and presence of IVH were not significant factors for unfavorable outcome or death in model I at certain ICH locations. Nevertheless, model I still had higher predictive performance than model II (ICH score alone). Elevated creatinine and glucose levels were independent predictors of death in patients due to lobar/putaminal hemorrhage. Hyperglycemia and diabetes mellitus were reported as independent predictors of poor outcomes in patients with mild-to-moderate ICH [30,31]. The effects of creatinine and glucose levels were not significant in other locations of hemorrhage, possibly due to the small samples of patients. Given that a wide range of GCS scores from 5 to 12 is assigned 1 point in the ICH score, the actual effect of differences in GCS score is difficult to trace. The ICH grading scale, which is a modified scale derived from the ICH score with more intervals for age, GCS score, and ICH volume, and has a total score from 5 to 13, may have higher sensitivity in predicting in-hospital and 30-day mortality than the ICH score does [17]. However, it may not be useful in identifying patients with a high probability of early death [32].

At least 37 prognostic grading scales have been developed to predict outcomes for patients with spontaneous ICH [33]. The modified ICH score, initially established by Cho et al. [18] for the assessment of basal ganglion hemorrhage, has not been equally valuable in the assessment of all ICH types [22]. The new modified ICH score uses the National Institute of Health Stroke Scale (NIHSS) instead of the GCS and achieved a slightly better prediction of good outcomes than the original ICH score did [34]. However, none of the scoring systems have stratified patients by hematoma location. Basal ganglion hemorrhage comprised the majority of ICH in all studied groups. Unless the numbers of patients in each ICH location are similar, bias can occur during weighing or scoring of the predictors from statistical analyses. Furthermore, scoring systems simply represent the disease severity or possible unfavorable outcomes and are not able to predict the probability of death. Considering the high risk of mortality, these quick simplified or convenient scores for death prediction are neither adequate nor satisfactory for clinical application. Prediction models by location-specific assessment systems with reliable nomograms provide better prediction performance for patient outcomes. In the present study, we developed location-specific nomograms with good discriminative ability and calibration for mortality. We could predict the mortality of patients by using the nomograms for appropriate ICH locations using the corresponding continuous variables.

For instance, a 79-year-old man (6.3 points) had a putaminal hemorrhage with an ICH volume of 65 cm^3^ (3.3 points), the presence of IVH (3 points), an initial GCS of 6 (7.6 points), a serum glucose level of 130 (2 points), and a creatinine level of 1.8 (1 point); this gave him a total score of 23.2 points, which corresponded to a probability of death of approximately 0.92 (92%; Figure 1A). With the ICH scoring system, a score of 3 was obtained, representing 30-day mortality of 26% in the study by Hemphill et al. [16] and a 55% predictive performance for death by the established nomogram using the ICH scoring system (Appendix A). Another example of the evaluation of a pontine hemorrhage through the nomogram for pontine hemorrhage in a 79-year-old woman (0 points) who had an ICH volume of 9 cm^3^ (2.6 points) with an initial GCS of 6 (7.6 points). Her total score of 10.2 points corresponded to a probability of death of 0.84 (84%). The ICH score was 2, which represented a 30-day mortality rate of only 13% in the study by Hemphill et al. and a 22% predictive performance for death by the established nomogram using the ICH scoring system (Appendix A). The wide variety of prediction probabilities indicates that the location-specific nomograms using continuous variables are better than the current scoring systems. The effect of surgical treatment was not included in the present study. With advancements such as minimally invasive surgical techniques and precision medicine, the outcomes of patients with ICH improve every year. The continual evolution of nomograms with updated information may be the ideal means of predicting patient outcomes.

This study had several limitations. First, this was a retrospective study. The number of patients, particularly those with thalamic, pontine, and cerebellar hemorrhages, was insufficient. The accuracy of nomograms developed using small samples of patients needs clarification. We speculate that a larger sample size will reveal age and the presence of IVH to be significant predictors of death due to all ICH types. Second, we did not perform an external validation of the nomograms. External validation with a different group of patients would help to improve the accuracy of the discrimination of the nomograms. Third, we did not use the NIHSS, which is not frequently used by neurosurgeons and explains functional status better than the GCS can in patients with ICH. The NIHSS represents overall neurological deficits, such as level of consciousness, degree of muscle power, speech, and swallowing functions, which correlate well with functional status. Further studies using the NIHSS instead of the GCS to establish nomograms will be more convincing. Fourth, we did not perform long-term follow-up; only short-term outcomes at discharge were available. Further longer-term prospective studies using these nomograms for clinical applications may provide more prognostic relevance. Finally, due to incomplete information, we did not analyze the significance of CRP. CRP was reported as being a potential independent predictor of ICH outcomes [35]. Regardless of these limitations, our data provide novel models of location-specific nomograms for more accurately predicting outcomes and mortality in patients with acute ICH.

## 5. Conclusions

Higher levels of immune-inflammatory markers are associated with larger ICH volume, worse initial GCS, and unfavorable outcome and death but are not independent prognostic predictors for death in patients due to acute ICH. Location-specific nomograms may be novel and accurate models for predicting mortality risk in patients with ICH.

## Figures and Tables

**Figure 1 diagnostics-12-00622-f001:**
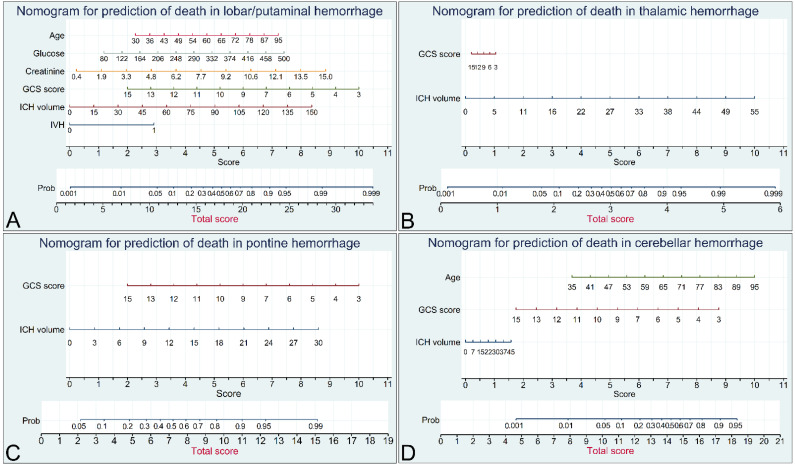
Nomograms for predicting death in patients due to lobar/putaminal (**A**), thalamic (**B**), ponine (**C**), and cerebellar hemorrhages (**D**). A vertical line is drawn from the value of each variable down to the “Score” line to match a score, and the matched scores are summed to obtain a total score. Then, a vertical line is drawn from the “Total Score” up to the “Prob” line to match the appropriate probability of death. GCS, Glasgow Coma Scale; ICH, intracerebral hemorrhage; IVH, intraventricular hemorrhage; Prob, probability.

**Figure 2 diagnostics-12-00622-f002:**
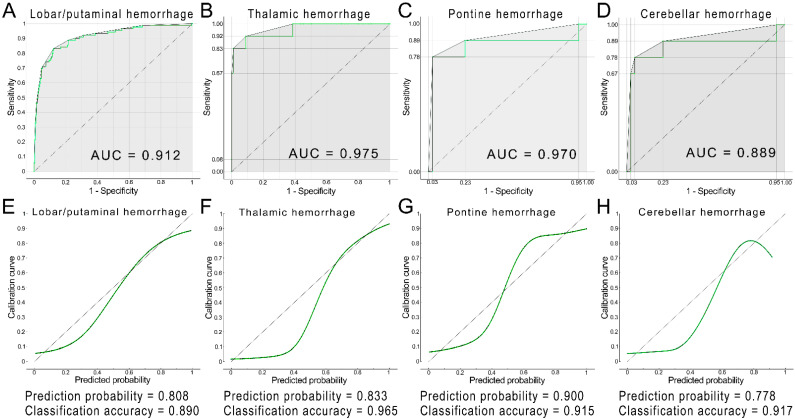
Area under the receiver operating characteristic curve of the nomogram for internal 5-fold cross-validation of death due to lobar/putaminal hemorrhage (**A**), thalamic hemorrhage (**B**), pontine hemorrhage (**C**), and cerebellar hemorrhage (**D**). Calibration curve of nomograms for the predicted probability of death due to lobar/putaminal hemorrhage (**E**), thalamic hemorrhage (**F**), pontine hemorrhage (**G**), and cerebellar hemorrhage (**H**). AUC, area under the curve.

**Table 1 diagnostics-12-00622-t001:** Baseline data of clinical features of 613 patients with acute intracerebral hemorrhage.

Characteristics	Total (*n* = 613)	Gender		*p*-Value *
		Men (*n* = 395)	Women (*n* = 218)	
Age (years)	64.0 (54.0–78.0)	61.0 (51.8–71.9)	70.8 (58.0–82.0)	<0.001
Systolic blood pressure (mmHg)	180 (153–207)	180 (154–205)	180 (151–209)	0.702
Diastolic blood pressure (mmHg)	97 (82–112)	100 (83–115)	93 (79–109)	0.001
Hemoglobin (g/dL)	14.1 (12.6–15.3)	14.7 (13.3–15.8)	13.2 (11.8–14.2)	<0.001
Platelet (×10^9^/L)	217 (171–263)	213 (170–256)	225 (179–269)	0.039
White blood cells (×10^3^/mL)	9.0 (6.9–11.7)	9.2 (7.0–11.9)	8.6 (6.7–11.5)	0.141
Neutrophil counts (×10^3^/mL)	6.1 (4.4–8.8)	6.3 (4.5–8.8)	5.9 (4.3–8.8)	0.352
Neutrophil-to-lymphocyte ratio	3.5 (2.0–7.0)	3.5 (2.0–6.9)	3.6 (2.0–7.3)	0.509
Systemic immune-inflammation index	746 (434–1548)	727 (436–1355)	784 (431–1827)	0.230
C-reactive protein (mg/L) ^a^	1.38 (0.37–6.21)	2.26 (0.41–6.91)	0.99 (0.29–3.35)	0.004
Glucose (mg/dL)	142 (117–174)	139 (114–169)	150 (121–185)	0.004
Creatinine (mg/dL)	1.0 (0.8–1.2)	1.1 (0.9–1.3)	0.8 (0.7–1.1)	<0.001
Initial Glasgow Coma Scale score	13 (8–15)	14 (9–15)	13 (7–15)	0.344
Hospital length of stay (days)	16 (7–25)	16 (7–25)	15 (15–25)	0.738
Modified Rankin Scale score	4 (3–5)	4 (3–5)	4 (3–5)	0.655
Hypertension	443 (72%)	282 (71%)	161 (74%)	0.572
Diabetes mellitus	138 (23%)	89 (23%)	49 (22%)	0.999
Heart disease	73 (12%)	40 (10%)	33 (15%)	0.069
Prior stroke	101 (16%)	65 (16%)	36 (17%)	0.999
Modified Rankin Scale score > 3	425 (69%)	272 (69%)	153 (73%)	0.784
Death	134 (22%)	88 (22%)	46 (21%)	0.760

Data are expressed as the median (1st–3rd quartile) or *n* (%). * Mann–Whitney test or chi-square test. ^a^ Available in 225 men and 126 women.

**Table 2 diagnostics-12-00622-t002:** Comparison of clinical features in 613 patients with intracerebral hemorrhage in different locations.

Characteristics	Lobe (*n* = 136)	Putamen (*n* = 238)	Thalamus (*n* = 113)	Caudate (*n* = 8)	Pons (*n* = 59)	Cerebellum (*n* = 47)	Ventricle (*n* = 12)	*p*-Value *
Age (years)	72.8 (60.6–83.6)	59.5 (49.8–72.9)	65.0 (58.0–77.7)	58.1 (52.5–75.0)	57.0 (49.3–70.9)	66.0 (60.0–72.9)	81.4 (58.5–88.4)	<0.001
Hemoglobin (g/dL)	13.3 (11.7–14.6)	14.3 (13.1–15.5)	14.4 (12.8–15.3)	14.7 (13.9–15.7)	14.8 (12.8–16.6)	14.0 (12.4–15.5)	13.7 (11.8–14.7)	<0.001
Platelet (×10^9^/L)	210 (165–253)	227 (180–268)	217 (166–263)	204 (174–229)	211 (170–262)	200 (173–261)	207 (161–275)	0.535
WBC (×10^3^/mL)	8.6 (6.6–11.5)	9.0 (7.0–11.7)	8.4 (6.4–10.7)	12.5 (7.9–16.0)	9.9 (7.2–13.0)	10.6 (7.4–13.7)	9.4 (7.4–10.7)	0.026
NC (×10^3^/mL)	5.9 (4.3–9.0)	6.1 (4.3–8.3)	5.7 (4.3–7.9)	11 (6.5–14.4)	6.7 (5.1–8.1)	7.7 (4.7–10.8)	7.2 (4.6–9.1)	0.029
SII	820 (440–1755)	672 (415–1267)	708 (469–1427)	1770 (1080–2691)	624 (280–1175)	891 (401–2588)	1451 (729–2047)	0.010
NLR	4.2 (2.2–7.7)	3.1 (1.8–6.4)	3.4 (2.1–6.2)	9.3 (5.9–19.7)	3.6 (1.7–5.4)	4.8 (2.0–12.3)	5.6 (3.1–8.9)	0.001
Glucose (mg/dL)	138 (117–164)	139 (115–167)	131 (111–169)	154 (147–163)	149 (128–176)	186 (156–237)	153 (123–181)	<0.001
Creatinine (mg/dL)	0.9 (0.7–1.2)	1.0 (0.8–1.2)	1.0 (0.8–1.2)	1.0 (0.9–1.1)	1.1 (0.8–1.5)	1 (0.8–1.5)	1.0 (0.9–1.1)	0.187
Hospital LOS (days)	16 (7–22)	16 (6–25)	18 (9–26)	20 (8–31)	8 (4–22)	16 (8–29)	17 (12–30)	0.132
ICH volume (cm^3^)	23 (9–40)	14 (6–40)	4 (2–7)	3 (2–3)	2 (1–7)	9 (2–24)	-	<0.001
Initial GCS score	14 (9–15)	13 (7–15)	14 (11–15)	14 (12–15)	11 (3–15)	14 (7–15)	15 (11–15)	0.005
Initial ICH score	1 (0–2)	1 (0–3)	1 (0–2)	1 (1–2)	2 (1–3)	2 (1–3)	-	<0.001
Discharge mRS score	4 (2–5)	4 (3–6)	4 (3–5)	4 (3–5)	5 (3–6)	4 (2–5)	4 (3–5)	0.104
Presence of IVH	24 (18%)	68 (29%)	49 (43%)	8 (100%)	10 (17%)	13 (28%)	12 (100%)	<0.001
mRS score > 3	84 (62%)	172 (72%)	81 (72%)	6 (75%)	43 (73%)	31 (66%)	8 (75%)	0.459
Death	27 (20%)	62 (26%)	12 (11%)	1 (13%)	21 (36%)	8 (17%)	3 (25%)	0.005

Data are expressed as the median (1st–3rd quartile) or *n* (%). * Kruskal–Wallis test or chi-square test. ICH, intracerebral hemorrhage; IVH, intraventricular hemorrhage; LOS, length of stay; mRS, Modified Rankin Scale; NC, neutrophil count; NLR, neutrophil-to-lymphocyte ratio; SII, systemic immune inflammation index.

**Table 3 diagnostics-12-00622-t003:** Linear regression analyses of correlation of age and immune-inflammation markers with clinical features in 613 patients with intracerebral hemorrhage.

Characteristics	Age			WBC			NC			NLR			SII		
	Coefficient	R^2^	*p*-Value	Coefficient	R^2^	*p*-Value	Coefficient	R^2^	*p*-Value	Coefficient	R^2^	*p*-Value	Coefficient	R^2^	*p*-Value
Age	-			−0.654	0.031	<0.001	−0.429	0.016	0.002	0.095	0.004	0.120	0.0001	0.000	0.643
Hb (g/dL)	−0.057	0.151	<0.001	0.115	0.043	<0.001	0.036	0.005	0.074	−0.023	0.011	0.011	−0.0005	0.004	0.138
Platelet (×10^9^/L)	−1.198	0.045	<0.001	7.159	0.115	<0.001	4.855	0.064	<0.001	0.025	0.000	0.942	-	-	-
Glucose (mg/dL)	−0.111	0.001	0.528	2.098	0.104	<0.001	3.4	0.057	<0.001	1.575	0.061	<0.001	0.007	0.068	<0.001
Cr (mg/dL)	−0.019	0.004	0.109	0.027	0.001	0.551	0.044	0.001	0.367	0.006	0.000	0.735	−0.0003	0.000	0.981
ICH volume (cm^3^)	0.201	0.012	0.007	2.251	0.109	<0.001	1.546	0.063	<0.001	0.338	0.015	0.003	0.005	0.021	<0.001
Initial GCS score	−0.026	0.008	0.023	−0.403	0.149	<0.001	−0.225	0.057	<0.001	−0.057	0.018	<0.001	−0.0002	0.017	0.001
Initial ICH score	0.025	0.076	<0.001	0.122	0.137	<0.001	0.075	0.062	<0.001	0.019	0.02	<0.001	0.0007	0.019	<0.001
mRS score	0.020	0.036	<0.001	0.105	0.067	<0.001	0.081	0.048	<0.001	0.026	0.025	<0.001	0.0009	0.0149	<0.001

Cr, creatinine; GCS, Glasgow Coma Scale; Hb, hemoglobin; ICH, intracerebral hemorrhage; mRS, Modified Rankin Scale; NC, neutrophil count; NLR, neutrophil-to-lymphocyte ratio; SII, systemic immune inflammation index; WBC, white blood cells.

**Table 4 diagnostics-12-00622-t004:** Univariate and multivariable analyses of clinical features and outcomes in 374 patients with lobar and putaminal hemorrhage.

Characteristics	Univariate Analysis						Multiple Logistic Regression	
	Unfavorable Outcome (mRS > 3)			Death			Death	
	No (*n* = 118)	Yes (*n* = 256)	*p*-Value *	No (*n* = 285)	Yes (*n* = 89)	*p*-Value *	OR (95% CI)	*p*-Value
Age (years)	62 (50.8–71.0)	66.4 (53.8–81.8)	0.004	63 (50.7–76.0)	69.0 (56.2–86.7)	<0.001	1.036 (1.010–1.063)	0.007
Hemoglobin (g/dL)	41.3 (13.4–15.3)	13.8 (12.0–15.1)	0.005	41.2 (12.6–15.2)	13.6 (11.4–14.7)	0.013	1.039 (0.856–1.262)	0.696
Platelet (×10^9^/L)	230 (197–265)	215 (168–263)	0.068	223 (178–265)	223 (223–255)	0.721		
WBC (×10^3^/mL)	8.5 (6.5–10.2)	9.1 (7.2–12.6)	0.004	8.4 (6.7–10.7)	10.6 (8.4–14.4)	<0.001	1.003 (0.902–1.115)	0.955
NC (×10^3^/mL)	5.6 (3.9–7.3)	6.2 (4.6–9.5)	0.010	5.7 (4.1–7.5)	7.7 (5.2–11.6)	<0.001		
NLR	2.9 (2.1–4.5)	3.8 (1.9–7.8)	0.037	3.3 (2.0–5.5)	4.9 (2.0–9.8)	0.009		
SII	676 (417–1091)	756 (432–1686)	0.193	678 (416–1229)	781 (464–2216)	0.069		
Glucose (mg/dL)	126 (108–145)	145 (120–175)	<0.001	131 (113–156)	165 (140–206)	<0.001	1.006 (0.999–1.013)	0.074
Creatinine (mg/dL)	0.9 (0.8–1.2)	1.0 (0.8–1.3)	0.027	1.0 (0.8–1.2)	1.1 (0.9–1.5)	0.006	1.313 (1.085–1.589)	0.005
Initial GCS score	15 (14–15)	10 (6–14)	<0.001	14 (11–15)	6 (3–8)	<0.001	0.767 (0.687–0.855)	<0.001
ICH volume (cm^3^)	6 (3–12)	27 (13–54)	<0.001	13 (5–26)	30 (32–93)	<0.001	1.024 (1.010–1.039)	<0.001
ICH score	0 (0–1)	2 (1–3)	<0.001	1 (0–1)	3 (3–4)	<0.001		
Presence of IVH	3 (2%)	89 (34%)	<0.001	33 (11%)	59 (66%)	<0.001	3.709 (1.635–8.412)	0.002
Female gender	39 (33%)	95 (37%)	0.487	102 (36%)	32 (36%)	>0.999		
Hypertension	82 (69%)	182 (71%)	0.807	201 (70%)	63 (70%)	>0.999		
Diabetes mellitus	23 (19%)	53 (20%)	0.890	58 (20%)	18 (20%)	>0.999		
Heart disease	13 (11%)	26 (10%)	0.856	33 (11%)	6 (6%)	0.236		
Prior stroke	10 (8%)	50 (19%)	0.006	40 (14%)	20 (22%)	0.069		

Data are expressed as the median (1st–3rd quartile) or *n* (%). * Mann–Whitney test or chi-square test. GCS, Glasgow Coma Scale; ICH, intracerebral hemorrhage; IVH, intraventricular hemorrhage; mRS, Modified Rankin Scale; LOS, length of stay; NC, neutrophil count; NLR, neutrophil-to-lymphocyte ratio; SII, systemic immune inflammation index; WBC, white blood cells.

**Table 5 diagnostics-12-00622-t005:** C-statistics of predictors of unfavorable outcomes and death for intracerebral hemorrhage in different locations.

Characteristics	Lobe/Putamen (*n* = 374)		Thalamus (*n* = 113)		Pons (*n* = 59)		Cerebellum (*n* = 48)	
	Unfavorable Outcome	Death	Unfavorable Outcome	Death	Unfavorable Outcome	Death	Unfavorable Outcome	Death
Model I (C-statistic)	0.876	0.931	0.832	0.993	0.9	0.959	0.846	0.944
Initial GCS score	+	+	+	+	+	+	+	+
ICH volume (cm^3^)	+	+	+	+	+	+	+	+
Presence of IVH	+	+						
Age (years)	+	+					+	+
Creatinine (mg/dL)	+	+						
Glucose (mg/dL)		+						
Model II (C-statistic)	0.824	0.893	0.717	0.937	0.853	0.885	0.818	0.94
ICH score	+	+	+	+	+	+	+	+
*p*-value *	<0.001	0.001	0.031	0.006	0.193	0.014	0.508	0.874

* Comparison of the C-statistic between models I and II. GCS, Glasgow Coma Scale; ICH, intracerebral hemorrhage; IVH, intra-ventricular hemorrhage. + indicates significant predictor.

## Data Availability

The data presented in this study are available on request from the corresponding author.

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
