# Peer review of "Correlation of Immune-Inflammatory Markers with Clinical Features and Novel Location-Specific Nomograms for Short-Term Outcomes in Patients with Intracerebral Hemorrhage"

_diagnostics, 2022, doi:10.3390/diagnostics12030622_

Round 1

Reviewer 1 Report

It is original research that contributes to the essential data about the  ICH risk factors and outcomes. It constitutes the basis for potential future therapeutic trials in clinical medicine.

The paper is well written, and a few minor shortcomings could be noticed:

36/ "Hemorrhagic stroke accounts for approximately 19% of acute strokes", it is true for Asian populations, for Caucasians the numbers are lower –According to Anderson CS et al. J Neurol Neurosurg Psych 1199457(10: 1173-9 or Go AS et al. Circulation 2014,21,129(3):399-410 and others  - about 71-87% of all strokes are ischemic strokes, 10-11% are intracerebral hemorrhage and 3-4 % are subarachnoid hemorrhages,  please correct it.

52/ "Most of the scores for ICH assessment are 52 based on age, level of consciousness, location, and size of the hemorrhage". Consider "score scales"?

63/ The aim is depicted  in a sophisticated  way and "multi-levels" – please put it more straightforward and with separate sentences, in its current form the text could confuse readers

117/ is: 2.4. Identification of hemorrhage location; should be: statistical methods or analysis

Author Response

Response to Reviewer 1 Comments

Manuscript ID: diagnostics-1607970

Title: Correlation of immune-inflammatory markers with clinical features and novel location-specific nomograms for short-term outcomes in patients with intracerebral hemorrhage

Thanks to reviewer’s precious comments. We have checked the manuscript and have made essential revisions according to reviewer’s comments point-by-point.

Point 1: "Hemorrhagic stroke accounts for approximately 19% of acute strokes", it is true for Asian populations, for Caucasians the numbers are lower –According to Anderson CS et al. J Neurol Neurosurg Psych 1199457(10: 1173-9 or Go AS et al. Circulation 2014,21,129(3):399-410 and others  - about 71-87% of all strokes are ischemic strokes, 10-11% are intracerebral hemorrhage and 3-4 % are subarachnoid hemorrhages,  please correct it.

Response: We have changed the statement to “Hemorrhagic stroke accounts for 8-15% in western countries and 18-24% in Asian countries, and carries a much higher risk of mortality than ischemic stroke does [1-4].” We have also added three new references (Ref. 1-3) here and have rearranged the number of the references.

Point 2: "Most of the scores for ICH assessment are based on age, level of consciousness, location, and size of the hemorrhage". Consider "score scales"?

Response: We have changed to “Most of the score scales for ICH…”.

Point 3: The aim is depicted  in a sophisticated  way and "multi-levels" – please put it more straightforward and with separate sentences, in its current form the text could confuse readers.

Response: We have changed the description of our aim as below: “Here, we investigated the association of the four immune-inflammatory markers—WBC count, NC, NLR, and SII—with clinical features and the outcomes of patients with ICH. We also sought to determine the appropriate criteria for different hematoma locations compared with the ICH score in predicting the outcomes of patients with ICH, and to establish nomograms to predict the probabilities of mortality at different ICH locations.”

Point 4: 2.4 Identification of hemorrhage location; should be: statistical methods or analysis.

Authors: We have changed to “2.5 Statistical Analysis”.

Reviewer 2 Report

In this study, the authors investigated the association of the four immune-inflammatory markers—WBC count, NC, NLR, and SII—with clinical features to determine the appropriate criteria for different hematoma locations compared with the ICH score in predicting the outcomes of patients with ICH. We also sought to establish nomograms to predict the probabilities of mortality at different ICH locations. This is a good clinical study. However, the number of enrolled patients is not large enough. Also, I was surprised to learn that CRP, a classic marker of ICH has not been evaluated (see, doi: 10.3389/fimmu.2018.01921)

Author Response

Response to Reviewer 2 Comments

Manuscript ID: diagnostics-1607970

Title: Correlation of immune-inflammatory markers with clinical features and novel location-specific nomograms for short-term outcomes in patients with intracerebral hemorrhage

Thanks to reviewer’s precious comments. We have checked the manuscript and have made essential revisions according to reviewer’s comments point-by-point.

Point 1: This is a good clinical study. However, the number of enrolled patients is not large enough. Also, I was surprised to learn that CRP, a classic marker of ICH has not been evaluated (see, doi: 10.3389/fimmu.2018.01921)

Response: As addressed in [Limitations]: “The number of patients, particularly those with thalamic, pontine, and cerebellar hemorrhage, was insufficient. The accuracy of nomograms developed using small samples of patients need clarification.” The CRP level was collected as well in this retrospective study. However, CRP was not a regular test for patients with ICH at the emergency department. The value of CRP was available only in 57% of patients. Therefore we did not include CRP for further analysis. We have addressed this point in both [Materials and Methods], [Results], Table 1, and [Limitations], and have added a new reference (Ref. 35) about the clinical evidence of CRP.
